# Development of a Real-Time qPCR Method for the Clinical Sample Detection of Capripox Virus

**DOI:** 10.3390/microorganisms11102476

**Published:** 2023-10-02

**Authors:** Jiaxin Wen, Xinying Yin, Xiaobo Zhang, Desong Lan, Junshan Liu, Xiaohui Song, Yu Sun, Jijuan Cao

**Affiliations:** 1Key Laboratory of Biotechnology and Bioresources Utilization of Ministry of Education, Dalian Minzu University, Dalian 116600, China; jx1069462542@163.com (J.W.); 18242096006@163.com (X.Y.); zhangxb2023@dlnu.edu.cn (X.Z.); 2Liaoning Center for Animal Disease Control and Prevention, Shenyang 110164, China; landesong@163.com; 3School of Mechanical Engineering, Faculty of Mechanical Engineering, Materials and Energy, Dalian University of Technology, Dalian 116024, China; liujs@dlut.edu.cn; 4China Animal Disease Prevention Control Center, Beijing 100125, China; songxiaohui0523@163.com

**Keywords:** Capripox virus (CaPV), detection, real-time quantitative PCR, clinical validation

## Abstract

Capripox viruses (CaPVs), including sheep pox virus (SPV), goat pox virus (GPV), and lumpy skin disease virus (LSDV), are the cause of sheep pox (SPP), goat pox (GTP), and lumpy skin disease (LSD) in cattle. These diseases are of great economic significance to farmers, as they are endemic on farms and are a major constraint to international trade in livestock and their products. Capripoxvirus (CaPV) infections produce similar symptoms in sheep and goats, and the three viruses cannot be distinguished serologically. In this study, we developed a real-time quantitative polymerase chain reaction (qPCR) method for identifying CaPV in goats, sheep, and cattle. Clinical samples were tested and verified. The developed assay was highly specific for target viruses, including GPVSPV and LSDV, which had no cross-reaction with other viruses causing similar clinical symptoms. An artificially synthesized positive control plasmid using the CaPV 32 gene inserted into the vector pMD19-T was used as a template, and the correlation coefficient of the linear regression curve (*R^2^*) was 0.9916, the estimated amplification efficiency (*E*) was 96.06%, and the sensitivity (limit of detection, LOD) was 3.80 copies per reaction. Using the clinical samples as a template, the limit of detection (LOD) was 4.91 × 10^−5^ ng per reaction (1.60 × 10^−5^–2.13 × 10^−3^ ng, 95% confidence interval (CI)), which means that this method was one of the most sensitive detection assays for CaPVs. A total of 85 clinical samples from CaPV-infected animals (goats, sheep, and cattle) and 50 clinical samples from healthy animals were used to test and compare the diagnostic results using the Synergy Brands (SYBR) Green-based PCR method recommended by the World Organization of Animal Health (WOAH). Both diagnostic sensitivity (*DSe*) (95.8–100%, 95% CI) and diagnostic specificity (*DSp*) (92.9–100%, 95% CI) results of the real-time quantitative PCR (qPCR) and SYBR Green PCR were 100%, and the kappa value (κ) was 1.0 (1-1, 95% CI). In summary, the assay established based on TaqMan probes was advantageous in high specificity, sensitivity, and general applicability and could be a competitive candidate tool for the diagnosis of CaPV in clinically suspected animals.

## 1. Introduction

Goat pox virus (GPV), sheep pox virus (SPV), and lumpy skin disease virus (LSDV) are three genetically similar pathogens of genus *Capripoxvirus* (CaPV) in the subfamily *Chordopoxvirinae*, family *Poxviridae* [1]. These viruses are the etiological agents of goat pox (GTP), sheep pox (SPP), and lumpy skin disease (LSD), respectively, which are acute viral infectious diseases of goats, sheep, and cattle [2]. Capripox (CaP) is endemic in many parts of the world, including China. It was initially found in Africa in 2012 and has since traveled through European countries and gradually spread to Asia; the epidemic is currently severe and is causing substantial economic losses in some countries in northern Africa, the Middle East, and Asia (Source: FAO, https://wahis.woah.org accessed on 24 September 2023) [3]. The outbreak of LSD was first confirmed in Xinjiang, China, in August 2019 [4]. At present, 24 cases of LSD have been reported in Fujian, Jiangxi, and other places in China [5]. There have also been 140 epidemic cases of GTP and SPP reported in Shanxi, Inner Mongolia, and other places in China, causing significant economic losses [6]. CaP affects the healthy development of the global ruminant breeding industry and the international trade in livestock products. It is also an important pathogen harming the ruminant breeding industry in China; accordingly, it is categorized as a notifiable disease by the World Organization for Animal Health (WOAH) [7].

LSD can infect cattle of all ages, both sexes, and all breeds. Its clinical symptoms include fever, depression, reduced milk yield, and skin lesions (nodules, papules, and spots) [8], with an incidence of 5–45% and a fatality rate of 10% [9]. The main clinical symptoms of goats and sheep infected with GTP and SPP are fever, erythema, papules, herpes, and blisters on the skin or mucosa of the hairless or lower parts [10,11], with an incidence of 50~80%; the mortality of adult sheep can reach 40%, and lamb mortality can reach 100% [12]. CaP is mainly transmitted via mechanical contact with blood-sucking arthropods such as mosquitoes, flies, and ticks [13]. Due to the diversity and non-controllability of mechanical transmissions of pests in the field, CaP has a significant influence on long-distance trans-regional diffusion. During detection, the disease can be preliminarily diagnosed through clinical symptoms such as fever, nodular changes in the skin, and lymph node enlargement, but its diagnosis depends on laboratory detection, especially when the disease is in the incubation period or prodromal stage [14]. Furthermore, the symptoms of skin damage in sick goats, sheep and cattle are easily confused with skin ringworm and insect bites. During the onset period, the infected nodules may present mucosal necrosis, rupture and other symptoms, which need to be differentiated from foot-and-mouth disease, peste des petits ruminants disease, bovine viral diarrhea, etc. [7,14,15]. In addition to judging by obvious clinical symptoms, GPV, SPV, and LSDV can also be diagnosed through virus isolation and culture [16], electron microscopy [17], animal experiments [18], and laboratory detection technology [19]. However, there may be mixed infections of multiple pathogens during the inoculation of disease materials, so it is necessary to combine molecular or serological detection methods with higher specificity to comprehensively determine whether the isolated pathogen is CaPV [20]. Moreover, silent infections without clinical symptoms also need to be confirmed by laboratory testing [21]. Laboratory diagnosis mainly relies on molecular biology and immunological detection methods [22]. As a highly contagious infectious disease, CaP has sporadic prevalence in China and neighboring countries and regions, and there is a risk of pandemic [1,23]. However, the ELISA method relies on antibodies, which have the limitations of popularization and application, and there is no commercial kit. At present, conventional PCR and SYBR Green PCR are the only molecular detection methods recommended by the WOAH which have the disadvantages of complicated operation. Therefore, the development of a rapid, accurate and simple CaPV-specific real-time qPCR detection method is a beneficial supplement to the recommended method of the WOAH, and also the key to the prevention and control of this disease.

In this study, a real-time qPCR method was established for the detection of CaPV, including GPV, SPV, and LSDV. The performance of the newly developed detection method was verified and compared with the SYBR Green-based PCR method recommended by the WOAH [24] in clinical samples to evaluate its degree of consistency.

## 2. Materials and Methods

### 2.1. Viruses, Bacteria and Specimens

All viruses and bacteria used for specificity analyses in this study were clinical samples or cultures, and they are listed in Table 1. Three members of the CaPVs genus (CaPV) (GPV, SPV and LSDV) were used. Other viruses and bacteria included foot-and-mouth disease virus (FMDV), peste des petits ruminants virus (PPRV), bovine viral diarrhea virus diarrhea virus (BVDV), Brucella, Mycobacterium tuberculosis, Bacillus anthracis. The clinical samples used in the study were from animal infection clinical samples. These samples were confirmed and stored by the Chinese Center for Animal Disease Control and Prevention (Beijing, China). Eighty-five clinical samples (skin tissue, whole blood, and lymph nodes) of CaPV infection were taken from goats (n = 26), sheep (n = 29), and cattle (n = 30), all showing mild to severe clinical symptoms (fever and skin lesions). The 50 negative control samples included samples from goats (n = 16), sheep (n = 15), and cattle (n = 19) and were collected from healthy animals. All clinical samples were confirmed using real-time SYBR Green PCR [8] as recommended by the WOAH in the Chinese Center for Animal Disease Control and Prevention. All clinical samples or cultures infected with viruses and bacteria were stored and used at the Chinese Center for Animal Disease Control and Prevention (Beijing, China).

### 2.2. Virus and Bacteria DNA/RNA Extraction

Clinical samples were used for method validation. For each tissue sample (skin tissue, lymph node), about 1.0 g was weighed from three different locations, cut and ground in the grinder, with 1.0 mL normal saline (0.9% sodium chloride solution) added before continuing grinding, and then homogenized and transferred to a sterile centrifuge tube, centrifuged in a high-speed refrigerated centrifuge at 10,000× *g* rpm for 2 min, and absorbed by using 100 µL supernatant in a 1.5 mL sterile centrifuge tube. The whole blood was centrifuged at 10,000× *g* rpm for 5 min, and 100 µL of supernatant was absorbed into a 1.5 mL sterile centrifuge tube. The supernatant, as described above, was stored at −80 °C for DNA extraction.

Viral and bacterial DNA (CaPV, Brucella, Mycobacterium tuberculosis, Bacillus anthracis) were extracted from clinical samples using the DNA extraction kit with the magnetic beads (Code DP438-T2K, Tiangen Biochemical Technology Co., Ltd., Beijing, China) according to the manufacturer’s instructions. Viral RNA (foot-and-mouth disease virus, peste des petits ruminants disease virus, bovine viral diarrhea virus) was extracted from clinical samples using the RNA extraction kit with the magnetic beads (Code DP452, Tiangen Biochemical Technology Co., Ltd., Beijing, China), according to the manufacturer’s instructions.

### 2.3. Homology Analysis

The gene sequences of GPV, SPV, and LSDV were retrieved from the National Center for Biotechnology Information (NCBI) nucleotide database (www.NCBI.nlm.nih.gov/nucleoties accessed on 24 September 2023). A total of 155 accession numbers (including 21 accession number LSDV isolates, 69 accession number SPV isolates, and 65 accession number GPV isolates) were searched based on the CaPV P32 gene sequence in NCBI. From 155 accession numbers, 39 representative isolates of 3 CaPV members (including different isolation times, locations, and hosts) were selected (Table 2). The CaPV P32 gene sequence was used to analyze the homology of the CaPV members (including GPV, SPV, and LSDV). The P32 gene sequence was used as a template in the blast search [25]. From the blast output, the results with the highest scores were selected in method development. In addition, gene sequences of other viruses and bacteria used for exclusivity tests were retrieved. MEGA 4.0 was used to compare gene sequences [26]. The 39 accession numbers of the P32 gene sequence of CaPV used in this study are shown in Table 2.

### 2.4. Primers and TaqMan Probes Design

The P32 gene sequence of CaPV and the gene sequences of closely related viruses and bacteria (including foot-and-mouth disease virus, peste des petits ruminants disease virus, bovine viral diarrhea virus, Brucella, Mycobacterium tuberculosis, and Bacillus anthracis) were subjected to multiple alignments, and the most volatile fragments were screened out. The 3’ end of the primer is placed at a location where the target virus displays different nucleotides from closely related viruses and bacteria, ensuring target virus specificity. Primers and probes were designed using Primer Express software 5.0 [27] (Applied Biosystems, Foster City, CA, USA), and secondary structures and the presence of possible primer dimers were evaluated. The nucleotide sequences used to design primer selection were introduced into the “Oligocalc” program [28], and the length was optimized according to the resulting “salt-adjusted” annealing temperature. The annealing temperature calculated by the “salt-adjusted” algorithm was used as the starting value for qPCR. The specific qPCR method for detecting the Capripox virus uses a TaqMan probe. The fluorescence group (Reporter) 6-carboxyfluorescein (6-FAM) was labeled at the 5’ end of the probe, and the Quencher group (Quencher) Black Hole Quencher 1(BHQ1) was labeled at the 3’ end of the probe. In addition, primers for the real-time SYBR Green PCR test (Fw3/Rev3) were derived from the WOAH Manual of Diagnostic Tests and Vaccines for Terrestrial Animals Chapter 3.8.12 on the diagnostic methods of Capripox [7]. Table 3 lists the sequence of all primers and probes used in this study. The primers and probes were synthesized by Takara Biomedical Technology (Dalian) Co., Ltd. (Dalian, China).

### 2.5. qPCR Reaction System

qPCR and SYBR Green PCR analyses were performed on an ABI Q7 Rapid thermal cycling apparatus (Thermo Fisher Scientific). Probe qPCR Mix (Code391A, TaKaRa Co., Ltd., Dalian, China) was used for the qPCR analysis. One-step PrimeScript III RT-qPCR Mix (RR600A, TaKaRa Co., Ltd., Dalian, China) was used for the RT-qPCR analysis. The composition of the PCR master mixture consisted of a 1 × kit supplement buffer (master mix) and 2 µL template (viral DNA or RNA), plus balanced nuclease-free water, resulting in a final volume of 25 µL. The final concentrations of primers and probes used in the PCR main mixture are as follows: 0.2 pmol/µL forward primer, 0.2 pmol/µL reverse primer, and 0.2 pmol/µL probe. The optimal thermal cycle conditions of qPCR (CaPV, Brucella, Mycobacterium tuberculosis, Bacillus anthracis) included 95 °C for 30 s followed by 1 cycle; 95 °C for 5 s, 60 °C for 30 s and 45 cycles. The optimized thermal cycle conditions of RT-qPCR (foot-and-mouth disease virus, peste des petits ruminants disease virus, bovine viral diarrhea virus) included 52 °C for 5 min (Reverse Transcript step), then 95 °C for 10 s, 95 °C for 5 s, 60 °C for 30 s and 45 cycles.

Real-time SYBR Green PCR, recommended by the WOAH, was used for the comparison, verification, and detection of the Capripoxvirus [29]. SYBR Green PCR analysis was performed using TB Green^®^ Premix Ex Taq™ II (Tli RNaseH Plus) (Code RR820A, TaKaRa Co., Ltd., Dalian, China). The optimal thermal cycle conditions of SYBR Green PCR (CaPV) included 95 °C for 15 min and 1 cycle, 95 °C for 15 s, 60 °C for 30 s, and 72 °C for 30 s and 45 cycles.

### 2.6. Artificial CaPV Templates Used as Positive Amplification Control (PAC)

The 550 bp sequence of the CaPV P32 gene (Accession no. MG458377.1) was inserted into the commercial vector pMD19-T to construct the plasmid, which was digested by Sac I and used as a linearized plasmid for positive amplification control (PAC). The synthesis of the PAC plasmid was completed by Sangon Biotech Co., Ltd. (Shanghai, China). The purified PAC plasmid was quantitatively determined using a NanoDrop 2000C spectrophotometer (Thermo Fisher Scientific) with a concentration of 34.45 ng/µL. The Avogadro number (6.022 × 10^23^ molecules/mole) was used to calculate the copy number, and the formula was as follows: copies/µL = [mass (g) × 6.022 × 10^23^(copies/mol)]/[length (bp) × 10^9^ × 650 (g/mol)]. The calculated copy number was 5.8 × 10^10^ copies/µL. PAC with copy number concentration was used as a template for detection limit (LOD) studies.

### 2.7. Amplification Efficiency (AE)

To calculate the amplification efficiency of the qPCR, seven series of dilution levels were prepared using the CaPV artificial positive amplification control (PAC) plasmid as a template, including 5.8 × 10^6^, 5.8 × 10^5^, 5.8 × 10^4^, 5.8 × 10^3^, 5.8 × 10^2^, 5.8 × 10^1^, 5.8 copies/µL (in copies). Each dilution level was analyzed 6 times and run under the condition of repeatability 4 times; the dilution series needed to be prepared again before each run. Each dilution level was measured 6 times, with 24 data points for each dilution point. The average Ct value obtained at each dilution level was plotted with PAC plasmid concentration, and linear regression analysis was performed using GraphPad Prism [21] (8.4.0, PRISM, Graphpad Software Inc., San Diego, CA, USA). Using the slope of the regression line, the formula AE = 100 (10^−1/slope^ − 1) was used to calculate the qPCR efficiency and was expressed as a percentage. For each target, the slope of the regression curve should be between −3.9 and −2.9, and the corresponding qPCR efficiency should range from 80% to 120%. Meanwhile, the correlation coefficient *R^2^* of the linear curve was an index to measure the linearity of the qPCR reaction. The *R^2^* of each target should be ≥0.98 [30].

### 2.8. Analytical Sensitivity

The analytical sensitivity of the qPCR method, which was presented by LOD_95%_, was determined by the experiment. The statistical significance LOD_95%_ was calculated using semi-logarithmic regression analysis (PRISM, Graphpad Software Inc., San Diego, CA, United States), with inputs of the corresponding number of sample materials, the number of repetitions, and the number of positive results in qPCR detection. In this study, the LOD_95%_ of two types of plasmid PACs (in copies of the genome) and the GPV clinical sample DNA template (in units of DNA concentration (ng/µL) were identified.

To determine the LOD_95%_ of CaPV using a qPCR-assay-based TaqMan probe, 8 series of dilution levels were prepared, including 5.8 × 10^4^, 5.8 × 10^3^, 5.8 × 10^2^, 5.8 × 10^1^, 5.8, 2.9, 0.58, 0.058 copies/µL, and the test was repeated at each dilution level not less than 8 times. The minimum number of copies per analysis was determined. In addition, the GPV clinical sample DNA was continuously diluted using healthy animal tissue DNA to prepare 9 series dilution levels as the template, including stock solution (10 ng/µL), 10^1^ × dilution (1 ng/µL), 10^2^ × dilution (10^−1^ ng/µL), 10^3^ × dilution (10^−2^ ng/µL), 10^4^ × dilution (10^−3^ ng/µL), 10^5^ × dilution (10^−4^ ng/µL), 10^6^ × dilution (10^−5^ ng/µL), 10^7^ × dilution (10^−6^ ng/µL), and 10^8^ × dilution (10^−7^ ng/µL). There were at least eight replicates per concentration. The maximum dilutive level for each clinical sample tested was determined.

### 2.9. Specificity and Cross-Reactivity Test

The specificity and cross-reactivity of the CaPV test assay were evaluated using qPCR. Undiluted sample material DNA/RNA was obtained using 9 closely related viruses and bacteria, including clinical samples or culture materials of differential viruses and bacteria of goat pox virus, sheep pox virus, lumpy skin disease virus, foot-and-mouth disease virus, peste des petits ruminants disease virus, bovine viral diarrhea virus, Brucella, Mycobacterium tuberculosis, and Bacillus anthracis, were tested using a CaPV exclusion assay based on the P32 gene. DNA/RNA analysis was repeated no less than 3 times for each sample material.

### 2.10. Diagnostic Sensitivity (DSe) and Diagnostic Specificity (DSp) Analysis

The diagnostic sensitivity (*DSe*) and specificity (*DSp*) of the qPCR assay were evaluated using clinical samples from naturally CaPV-infected animals (goats, sheep, cattle) and healthy animals. The diagnostic performance was compared with the SYBR Green qPCR method recommended by the WOAH [22], and the diagnostic performance of the qPCR analytical method developed in this study in clinical samples was discussed. By calculating the kappa value, the consistency of the validation of clinical samples was analyzed. Eighty-five clinical samples taken from CaPV-infected animals (goats, sheep, cattle) were used, including 20 skin tissue samples, 46 whole blood samples, and 19 lymph node samples. A total of 50 clinical samples were taken from healthy animals (goats, sheep, cattle), including 9 skin tissue samples, 33 whole blood samples, and 8 lymph node samples. Probabilistic regression analysis was performed using MedCalc Software bvba (Ostend, Belgium) [31] to calculate the degree of consistency of this test against clinical samples at the 95% probability level.

## 3. Results

### 3.1. Development of the Specific qPCR Method for CaPV

We selected the P32 gene of CaPV as the target developed using the real-time qPCR method for CaPV detection. Three member isolates of CaPV (including GPV, SPV, and LSDV) with thirty-nine representative entry numbers were selected (Table 2) to analyze the P32 gene homology of these isolates. For the P32 gene sequence, MEGA 4.0 was used to conduct the homology alignment of the primer and probe sequences (Figure 1). The sheep pox virus (CaPV) specific primer-probe designed in this study basically matched the sheep pox virus (CaPV) gene sequence of 39 entry numbers and could simultaneously detect the three serotypes of CaPV, including GPV, SPV, and LSDV. In addition, six closely related and different viruses and bacteria (including foot-and-mouth disease virus, peste des petits ruminants virus, bovine viral diarrhea virus, Brucella, Mycobacterium tuberculosis, and Bacillus anthracis) were retrieved in NCBI, and gene sequences were compared using MEGA 4.0. The sequences of the forward primers, TaqMan probes, and reverse primers of CaPV basically did not match those of different viruses and bacteria. Theoretically, CaPV can be specifically detected, and other closely related different viruses and bacteria can be excluded (Figure 1).

### 3.2. Amplification Efficiency (AE)

To evaluate the qPCR efficiency of this method, seven series of dilution levels were prepared using the CaPV artificial positive amplification control (PAC) plasmid as a template, including 5.8 × 10^6^, 5.8 × 10^5^, 5.8 × 10^4^, 5.8 × 10^3^, 5.8 × 10^2^, 5.8 × 10^1^, and 5.8 copies/µL, and they were analyzed using TaqMan probe qPCR. A linear regression curve was plotted between the threshold cycle value (Ct value) and PAC plasmid concentration (in copy number). According to the equation *E* = 100 (10^−1/slope^ − 1), the analysis efficiency was determined as the slope of the regression line, showing a good linear relationship between the Ct value and PAC plasmid concentration (Figure 2). The correlation coefficient *R^2^* of the real-time qPCR method for CaPV detection was 0.9916, the slope of the regression curve was −3.42, and the efficiency was 96.06%. These results meet the requirements specified in the general qPCR validation guidelines, i.e., correlation coefficient *R^2^* ≥ 0.98, regression curve slope from −3.9 to −2.9, and efficiency of 80–120%.

### 3.3. Analytical Sensitivity Using PACs as a Template

The LOD_95%_ of PACs were expressed as the copy number per analysis, where each copy represents one genome copy of the virus particle. Eight series of dilution levels were prepared using the CaPV artificial positive amplification control (PAC) plasmid as a template, including 5.8 × 10^4^, 5.8 × 10^3^, 5.8 × 10^2^, 5.8 × 10^1^, 5.8, 2.9, 0.58, and 0.058 copies/µL (eight replicates for per concentration), to evaluate the LOD_95%_ of CaPV using a real-time qPCR assay. The expansion of PACs was linear. The results of probabilistic regression analysis showed that the LOD_95%_ of CaPV using the qPCR assay was 3.80 copies/response, with a 95% confidence interval of 1.88–38.19 copies/response (Figure 3a).

### 3.4. Analytical Sensitivity Using Clinical Sample DNA as a Template

Using clinical sample DNA as a template, the LOD_95%_ was expressed as the highest dilutive level for each test. Using the 10 × dilution GPV DNA samples of healthy goat skin tissue samples as the template, we prepared 10 series of dilution levels, including stock solution (10 ng/µL), 10^1^ × dilution (1 ng/µL), 10^2^ × dilution (10^−1^ ng/µL), 10^3^ × dilution (10^−2^ ng/µL), 10^4^ × dilution (10^−3^ ng/µL), 10^5^ × dilution (10^−4^ ng/µL), 10^6^ × dilution (10^−5^ ng/µL), 10^7^ × dilution (10^−6^ ng/µL), 10^8^ × dilution (10^−7^ ng/µL), and 10^9^ × dilution (10^−8^ ng/µL), as templates (eight replicates per concentration) to evaluate the LOD_95%_ of CaPV using the qPCR assay—the amplification using the clinical sample DNA as the template was linear. The results of the probabilistic regression analysis show that the LOD_95%_ of CaPV determined using the qPCR assay was 4.91 × 10^−5^ ng/µL (i.e., the highest dilution level of each test was the 10^6^ × dilution level). The 95% confidence interval was 1.60 × 10^−5^–2.13 × 10^−3^ ng (Figure 3b).

### 3.5. Specificity and Cross-Reactivity

To determine the specificity of the method, clinical samples of infection from three members of CaPV (GPV, SPV, and LSDV) and samples from six non-targeted, undifferentiated viruses and bacteria (including foot-and-mouth disease virus, peste des petits ruminants disease virus, bovine viral diarrhea virus, Brucella, Mycobacterium tuberculosis, and Bacillus anthracis) were used. The specificity of the real-time qPCR assay for CaPV was evaluated (Table 4 and Figure 4). Exclusion test results based on real-time qPCR showed that the specificity for three members of CaPV (GPV, SPV, and LSDV) was 100%, and there was no cross-reaction against six different non-targeted viruses and bacteria. These results confirmed the accuracy and specificity of the detection method.

### 3.6. Diagnostic Sensitivity (DSe) and Diagnostic Specificity (DSp) of Clinical Validation

The diagnostic sensitivity (*DSe*) and diagnostic specificity (*DSp*) of the real-time qPCR method for the diagnosis of CaPV were calculated using probability regression analysis. Samples from 85 CaPV-infected animals (goats, sheep, and cattle) and 50 healthy animals (goats, sheep, and cattle) were used for clinical verification. These clinical samples were confirmed using the SYBR Green qPCR method recommended by the WOAH.

Using the real-time qPCR method and the SYBR Green qPCR method recommended by the WOAH, 26 GPV-infected goats (with samples including six skin tissues, fourteen whole blood, and six lymph nodes), 29 SPV-infected sheep (with samples including six skin tissues, sixteen whole blood, and seven lymph nodes) and 30 LSDV-infected cattle (with samples including eight skin tissues, sixteen whole blood and six lymph nodes) were clinically validated, and all tests were positive (Table 5). We have arranged and summarized the Ct values of these CaPV-infected samples from small to large based on qPCR-based detection in Table 5. By comparing the Ct values, it was found that the qPCR Ct values of the 26 GPV-infected goats ranged from 15.36 ± 0.16 to 27.41 ± 0.09, and the qPCR Ct values of SYBR Green ranged from 14.29 ± 0.09 to 27.65 ± 0.04 (Figure 5A). The Ct values of 29 SPV-infected sheep ranged from 13.34 ± 0.00 to 28.36 ± 0.01 and 10.18 ± 0.09 to 22.69 ± 0.06 (Figure 5B). The Ct values of 30 LSDV-infected cattle ranged from 12.44 ± 0.14 to 27.34 ± 0.04 and 10.07 ± 0.07 to 32.69 ± 0.06 (Figure 5C). These two methods showed a good correlation of the Ct values. It was observed that the real-time qPCR method showed better detection sensitivity from a sample of mildly infected CaPV (LG30), and we need to collect more samples of mildly infected CaPV to confirm this finding.

In total, 50 samples of healthy animal materials (including 16 healthy goats, 15 healthy sheep, and 19 healthy cattle) were clinically verified, and all tests were negative (Table 5). At the same time, the SYBR Green PCR method recommended by the WOAH was used to compare the specificity and accuracy of the methods. The diagnostic sensitivity (*DSe*) and specificity (*DSp*) of the real-time qPCR method for CaPV clinical samples were 100% (95.8~100%, 95% CI) and 100% (92.9~100%, 95% CI), respectively. Kappa values were 1.0 (1-1, 95% CI) (Table 6).

## 4. Discussion

In this study, a real-time qPCR assay was developed to detect GPV, SPV, and LSDV simultaneously and to exclude multiple other pathogens from samples containing multiple pathogens.

At the beginning of development, we considered the coverage of CaPV isolates worldwide in terms of sequence retrieval and the design of primer probes. Since the P32 protein of CaPV has high sequence homology among GPV, SPV, and LSDV, it is a common structural protein with high specificity and strong immunogenicity in CaPV isolates [32,33]. Therefore, 155 accession numbers of the CaPV P32 protein gene were searched inNCBI, and 39 representative isolates of three CaPV members (GPV, SPV, LSDV) were selected from them, including different isolation times, different isolation sites, and different hosts to analyze the homology of the P32 protein gene for designing primers and probes. In addition, we also investigated the amplification efficiency of the qPCR method, which was 96.06%. Others also reported the development of detection methods for three CaPV members [10,34,35]. These methods were not used to analyze the isolation time, location, and host conditions of CaPV isolates at the time of development and were used to investigate the amplification efficiency. Ruminants and their products are circulated worldwide, which increases the risk of the CaP spreading across borders. It is beneficial to ensure the accuracy of CaPV detection to develop the qPCR method and evaluate the amplification efficiency based on sufficient isolation time, isolation location, and the analysis of CaPV isolates in the host.

In terms of sensitivity (LOD), this study used continuous diluents of DNA from clinical samples of CaPV and plasmid PACs as templates to investigate the LOD_95%_ of two types, respectively. Probabilistic regression analysis shows that for the assay of qPCR, the LOD_95%_ was 3.80 copies per response (1.88–38.19 copies per response, 95% CI), 4.91 × 10^−5^ ng per response (1.60 × 10^−5^–2.13 × 10^−3^ ng, 95% CI). The maximum dilution level of DNA detected in clinical samples was a 10^6^ × dilution level. Others also reported that the LOD of the CaPV detection method reached 10 copies [22,32,35]. The highest dilution level of the DNA template in clinical samples was examined, and it was more similar to the situation of mild CaPV infection in animals.

The CaPV exclusion test showed 100% specificity for all serotypes/thesolates of the target viruses tested (GPV, SPV, and LSDV) and did not cross-react with clinical samples or culture materials of other viruses and bacteria. The *dSe* evaluation of 135 clinical samples showed 100% (95.8% to 100%, 95% CI) *dSe* and 100% (92.9% to 100%, 95% CI) *dSe* and *DSp* compared to the SYBR Green PCR recommended by the WOAH. The Kappa value of 1.0 (1-1, 95% CI) indicated that the two methods are comparable.

Capripox (CaP) is a major infectious disease affecting the healthy development of the ruminant breeding industry in China [11]. Currently, there is no effective drug treatment, and comprehensive prevention and control measures such as vaccine immunization, pathogen and antibody monitoring, and population purification are the main responses. According to the standard of the WOAH Manual of Diagnostic Tests and Vaccines for Terrestrial Animals, Chapter 3.8.12, on the diagnostic methods of Capripox [29], these include virus isolation, electron microscope examination, inclusion body examination, neutralization tests, PCR method, etc. These methods have complex operations, long test cycles, and low sensitivity. The real-time qPCR method developed in this study can be a good supplement for the early detection of CaP disease in ruminants.

## 5. Conclusions

The study describes a real-time quantitative PCR method based on a TaqMan probe for identifying CaPV in goats, sheep, and cattle and its validation in clinical samples. The amplification efficiency (*E*) of the method was 96.06%. The LOD_95%_ of serial dilutions of DNA from plasmid PACs and CaPV clinical samples were 3.80 copies per reaction (1.88–38.19 copies per reaction, 95% CI) and 4.91 × 10^−5^ ng per reaction (1.60 × 10^−5^–2.13 × 10^−3^ ng, 95% CI). The highest dilution level for detecting DNA in clinical samples was the 10^6^ × dilution level. Non-targeted differential viruses and bacteria were not detected using this method. Compared with the SYBR Green PCR recommended by the WOAH, the diagnostic sensitivity and specificity were both 100% (95.8–100%, 95%CI; 92.9–100%, 95%CI). The κ value was 1.0 (1-1, 95% CI).

In summary, the established qPCR method is highly sensitive, specific, and reproducible and can be used as an effective tool for detecting CaP disease in ruminants early.

## Figures and Tables

**Figure 1 microorganisms-11-02476-f001:**
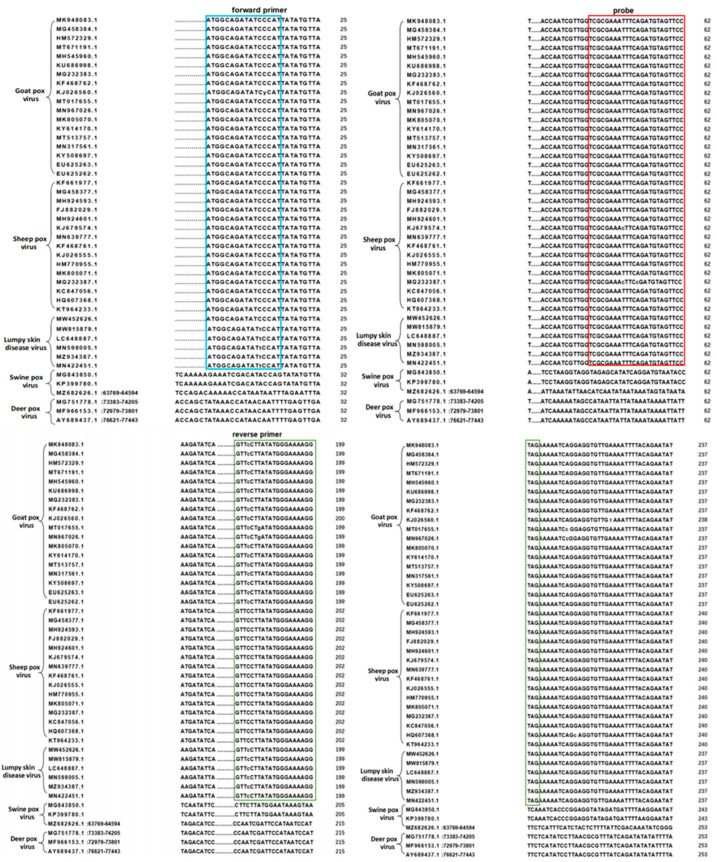
A sequence of P32 gene regions of different species developed for CaPV-specific qPCR analysis. The positions of forward primers, reverse primers, and probes are indicated by blue, green, and red boxes, respectively.

**Figure 2 microorganisms-11-02476-f002:**
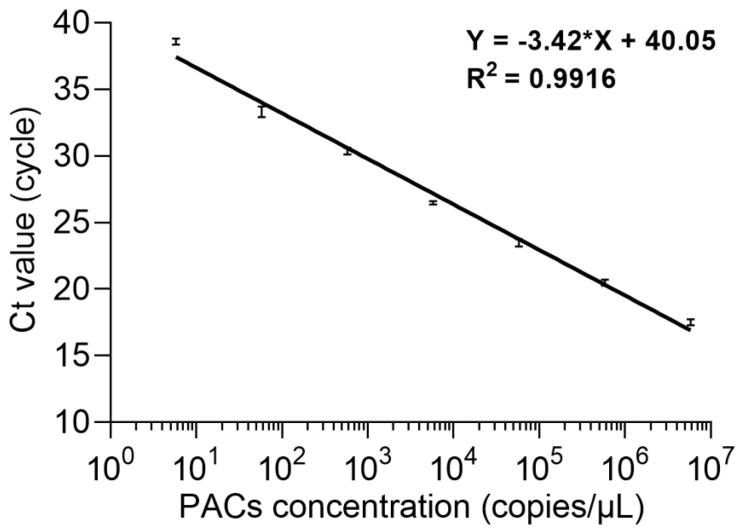
Standard curves of the PCAs analyses of seven dilution levels for the real-time qPCR for CaPV. *E* = 96.06%; slope = −3.42; and *R^2^* = 0.9916.

**Figure 3 microorganisms-11-02476-f003:**
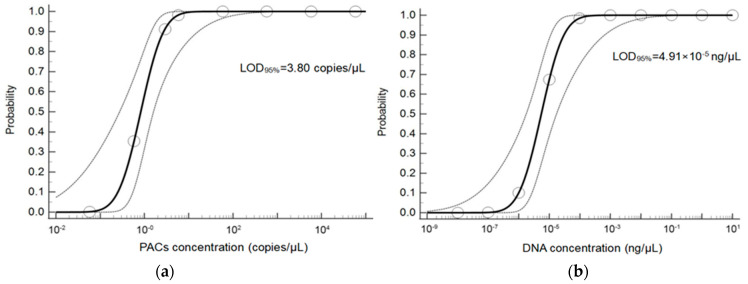
Sensitivity analysis results. (**a**) The LOD_95%_ of sensitivity for PACs as a template. Probit regression analysis using MedCalc software was performed on data from 8 replicates from serial dilutions using the qPCR for CaPV; (**b**) the LOD_95%_ of sensitivity for clinical sample DNA as a template. Probit regression analysis using MedCalc software was performed on data from 8 replicates from ten serial dilutions using the qPCR for CaPV.

**Figure 4 microorganisms-11-02476-f004:**
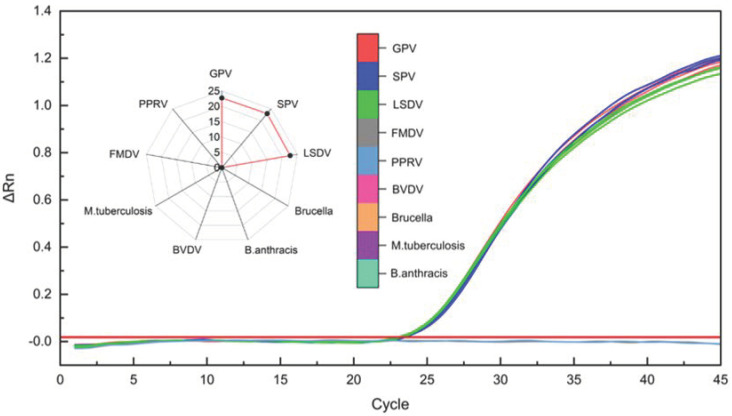
Specificity and cross-reactivity of the TaqMan probe qPCR assay.

**Figure 5 microorganisms-11-02476-f005:**
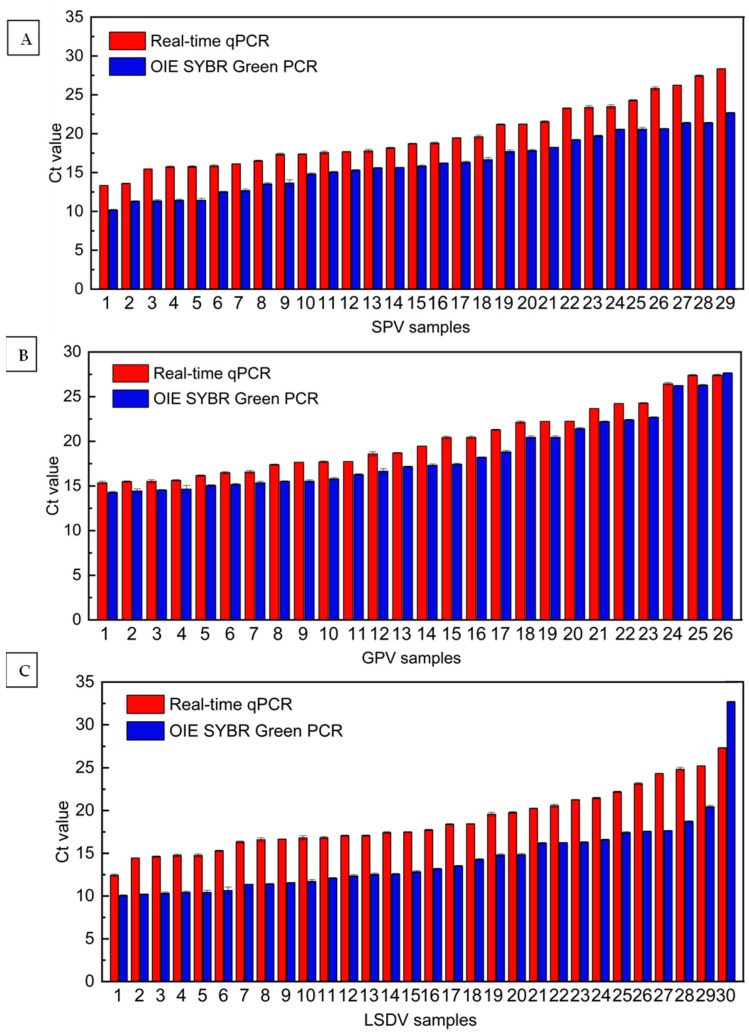
Real-time qPCR and SYBR Green qPCR were used to verify 85 CaPV-infected animals. Samples from 26 GPV-infected goats (**A**), 29 SPV-infected sheep (**B**), and 30 LSDV-infected cattle (**C**).

**Table 1 microorganisms-11-02476-t001:** List of clinical samples used for method development specificity analysis and their sources.

Viruses, Bacteria, and Specimens	Genotype	Sample Infection Type	Isolation Times/Locations	Number of Specimens
Capripoxvirus	GPV	Clinical samples of goat infection (natural hosts)	2023/Xinjiang	2023/3/CaPV-G
	SPV	Clinical samples of sheep infection (natural hosts)	2023/Xinjiang	2023/3/CaPV-S
	LSDV	Clinical samples of cattle infection (natural hosts)	2023/Xinjiang	2023/3/CaPV-L
Foot-and-mouth disease virus		Clinical samples of cattle infection (natural hosts)	2023/Xilingol League, Inner Mongolia	2023/3/FMDV
Peste des petits ruminants virus		Inactivated vaccine (natural hosts)	2023/Purchase from HUAPAI BIOLOGICAL GROUP	2023/3/PPRV
Bovine viral diarrhea virus		Clinical samples of cattle infection (natural hosts)	2023/Baragaer Gol township, Xiwuqi,	2023/3/BVDV
Brucella		Inactivated vaccine (natural hosts)	2023/Purchase from AOLONG BIOLOGICAL GROUP	2023/3/Bru
Mycobacterium tuberculosis		Clinical samples of cattle infection (natural hosts)	2023/Xilingol League, Inner Mongolia	2023/3/M.tube
Bacillus anthracis		Inactivated vaccine (natural hosts)	2023/Purchase from HAYAO BIOLOGICAL GROUP	2023/3/Bac.

**Table 2 microorganisms-11-02476-t002:** The 39 accession numbers of the P32 gene of Capripox virus (CaPV) used for real-time qPCR assays from NCBI.

Species	Isolates	Accession No.	Isolation Times/Locations	Host
GPV	Gorgan	MK948083.1	2019/Iran	Goat
GPV	ShanX-YA	MG458384.1	2019/China	Goat
GPV	GPV/ChongQ	HM572329.1	2011/China	Goat
GPV	P32/Menasagere/Mandya	MT671191.1	2021/India	Goat
GPV	APET/KPM/TN/16	MH545960.1	2019/India	Goat
GPV	GPV4/WB/10	KU686998.1	2016/India	Goat
GPV	GTPV/SA2	MG232383.1	2018/Saudi Arabia	Goat
GPV	Maha/goat/19	KF468762.1	2015/India	Goat
GPV	Zl/Gs	KJ026560.1	2014/China	Goat
GPV	Domestic goat/Tawang(AR)	MT017655.1	2020/India	Domestic goat
GPV	Goral/AR/2018	MN967026.1	2020/India	Naemorhedus goral
GPV	Puducherry-216/NIVEDI	MK805070.1	2019/India	Goat
GPV	GTPV12/WB/10	KY614170.1	2018/India	Goat
GPV	NIVEDI/Kandhnahalli/Chellakere	MT513757.1	2021/India	Goat
GPV	VNUAGTP1	MN317561.1	2020/Vietnam	Goat
GPV	GPV 28/TN/15	KY508697.1	2017/India	Goat
GPV	Vietnam (Ninh Thuan/2005) IMV	EU625263.1	2008/Vietnam	Goat
GPV	Yemen (Sana’a/1983) IMV	EU625262.1	2008/Yemen	Goat
SPV	GanSuHN/12	KF661977.1	2013/China	Sheep
SPV	QinH-HD	MG458377.1	2019/China	Sheep
SPV	36/16	MH924593.1	2018/Tunisia	Sheep
SPV	Pune-08	FJ882029.1	2009/India	Sheep
SPV	1517/12M	MH924601.1	2018/Tunisia	Sheep
SPV	RF	KJ679574.1	2016/India	Sheep
SPV	Kanakapura-NI/NIVEDI	MN639777.1	2020/India	Sheep
SPV	Maha/sheep/22	KF468761.1	2015/India	Sheep
SPV	Lx/Gs	KJ026555.1	2014/China	Sheep
SPV	Shanxi	HM770955.1	2010/China	Sheep
SPV	JK-221/NIVEDI	MK805071.1	2019/India	Sheep
SPV	SPPV/SA6/2016	MG232387.1	2018/Saudi Arabia	Sheep
SPV	Zabaikalsk	KC847056.1	2014/Russia	Sheep
SPV	AV40	HQ607368.1	2011/China	Sheep
SPV	Envelope protein	KT964233.1	2016/Tunisia	Sheep
LSDV	IND/WB/JS10-LT	MW452626.1	2021/India	Cattle
LSDV	RIVER/VMC/LSDV/01/Puducherry	MW815879.1	2022/India	Cattle
LSDV	LSVN/2020	LC648887.1	2021/Vietnam	Bos taurus
LSDV	Xinjiang/2019	MN598005.1	2020/China	Cattle
LSDV	KM/Taiwan/2020	MZ934387.1	2022/China Taiwan	Cattle
LSDV	KSA6/2017	MN422451.1	2020/Saudi Arabia	Cattle

**Table 3 microorganisms-11-02476-t003:** Sequence information of primers and probes used for PCR analysis in this study.

Viruses	Primer	Sequence (5′–3′)	Genbank No.	Position	Size (bp)	Reference
Capripoxvirus	Fw1	ATGGCAGATATC(t)CCATTA	MG458377.1	1–18	209	This work
	Rev1	CTACCTTTTCCCATATA(c)AGT(c)AAC		187–209		
	Probe	FAM-TCGCGAAATTTCAGATGTAGTTCCA-BHQ1		48–73		
Capripoxvirus	Fw3	TGGGAAAAGGTAGAAAAATCAGGAGG	MG458377.1	197–221	141	[29]
	Rev3	ATCCGCATCGGCATACGATT		318–337		

**Table 4 microorganisms-11-02476-t004:** Specificity analysis results from the TaqMan probe qPCR assay.

Virus and bacteria	Sample Type	Average Ct Value
GPV	Clinical samples of goat infection	22.71 ± 0.38
SPV	Clinical samples of sheep infection	22.99 ± 0.47
LSDV	Clinical samples of cattle infection	22.61 ± 0.24
Foot-and-mouth disease virus	Clinical samples of cattle infection	Negative
Peste des petits ruminants virus	Inactivated vaccine	Negative
Bovine viral diarrhea virus	Clinical samples of cattle infection	Negative
Brucella	Inactivated vaccine	Negative
Mycobacterium tuberculosis	Clinical samples of cattle infection	Negative
Bacillus anthracis	Inactivated vaccine	Negative

**Table 5 microorganisms-11-02476-t005:** Validation of 135 clinical samples using real-time PCR and SYBR-Green-based PCR.

Samples	Type	TaqMan Probe qPCR	WOAH SYBR Green PCR
		Ct Value	Ct Value
GG1	GPV skin tissue	17.39 ± 0.06	15.18 ± 0.09
GG2	GPV skin tissue	20.44 ± 0.13	17.43 ± 0.11
GG3	GPV skin tissue	22.14 ± 0.13	22.69 ± 0.06
GG4	GPV skin tissue	22.25 ± 0.02	20.45 ± 0.16
GG5	GPV skin tissue	24.28 ± 0.06	16.66 ± 0.28
GG6	GPV skin tissue	27.41 ± 0.09	16.29 ± 0.08
GG7	GPV whole blood	15.36 ± 0.16	14.56 ± 0.04
GG8	GPV whole blood	15.50 ± 0.07	14.65 ± 0.41
GG9	GPV whole blood	15.52 ± 0.18	15.36 ± 0.16
GG10	GPV whole blood	15.63 ± 0.08	14.44 ± 0.25
GG11	GPV whole blood	16.17 ± 0.07	15.52 ± 0.06
GG12	GPV whole blood	16.57 ± 0.17	14.29 ± 0.09
GG13	GPV whole blood	17.65 ± 0.01	15.07 ± 0.07
GG14	GPV whole blood	17.74 ± 0.01	15.54 ± 0.15
GG15	GPV whole blood	18.71 ± 0.05	17.33 ± 0.14
GG16	GPV whole blood	19.46 ± 0.02	18.83 ± 0.14
GG17	GPV whole blood	21.29 ± 0.06	26.25 ± 0.01
GG18	GPV whole blood	23.68 ± 0.01	22.21 ± 0.06
GG19	GPV whole blood	24.24 ± 0.01	22.40 ± 0.07
GG20	GPV whole blood	26.45 ± 0.14	27.65 ± 0.04
GG21	GPV lymph node	16.50 ± 0.10	17.17 ± 0.06
GG22	GPV lymph node	17.71 ± 0.09	18.22 ± 0.04
GG23	GPV lymph node	18.60 ± 0.23	15.81 ± 0.14
GG24	GPV lymph node	20.44 ± 0.13	21.43 ± 0.11
GG25	GPV lymph node	22.24 ± 0.01	20.45 ± 0.16
GG26	GPV lymph node	27.40 ± 0.08	26.29 ± 0.08
SG1	SPV skin tissue	13.34 ± 0.00	11.43 ± 0.14
SG2	SPV skin tissue	15.46 ± 0.01	11.29 ± 0.08
SG3	SPV skin tissue	15.75 ± 0.11	12.68 ± 0.20
SG4	SPV skin tissue	17.38 ± 0.05	10.18 ± 0.09
SG5	SPV skin tissue	23.29 ± 0.05	16.66 ± 0.28
SG6	SPV skin tissue	27.45 ± 0.13	20.58 ± 0.21
SG7	SPV whole blood	15.73 ± 0.11	11.36 ± 0.16
SG8	SPV whole blood	15.85 ± 0.16	13.54 ± 0.15
SG9	SPV whole blood	16.12 ± 0.00	12.52 ± 0.06
SG10	SPV whole blood	16.51 ± 0.08	13.65 ± 0.41
SG11	SPV whole blood	17.58 ± 0.18	15.84 ± 0.14
SG22	SPV whole blood	17.67 ± 0.04	15.07 ± 0.07
SG13	SPV whole blood	18.17 ± 0.08	15.60 ± 0.06
SG14	SPV whole blood	18.71 ± 0.06	16.33 ± 0.14
SG15	SPV whole blood	18.79 ± 0.13	15.29 ± 0.09
SG16	SPV whole blood	19.45 ± 0.04	17.83 ± 0.14
SG17	SPV whole blood	21.24 ± 0.01	17.72 ± 0.21
SG18	SPV whole blood	21.55 ± 0.13	19.72 ± 0.10
SG19	SPV whole blood	23.39 ± 0.22	22.69 ± 0.06
SG20	SPV whole blood	23.49 ± 0.23	20.56 ± 0.05
SG21	SPV whole blood	25.82 ± 0.21	19.21 ± 0.06
SG22	SPV whole blood	28.36 ± 0.01	21.39 ± 0.09
SG23	SPV lymph node	13.60 ± 0.04	11.44 ± 0.25
SG24	SPV lymph node	17.35 ± 0.15	15.65 ± 0.04
SG25	SPV lymph node	17.79 ± 0.20	16.22 ± 0.04
SG26	SPV lymph node	19.60 ± 0.23	14.81 ± 0.14
SG27	SPV lymph node	21.19 ± 0.07	18.25 ± 0.01
SG28	SPV lymph node	24.29 ± 0.08	20.65 ± 0.04
SG29	SPV lymph node	26.24 ± 0.01	21.40 ± 0.07
LG1	LSDV skin tissue	12.44 ± 0.14	10.44 ± 0.13
LG2	LSDV skin tissue	14.45 ± 0.01	12.09 ± 0.07
LG3	LSDV skin tissue	14.77 ± 0.14	12.57 ± 0.07
LG4	LSDV skin tissue	17.43 ± 0.11	13.18 ± 0.07
LG5	LSDV skin tissue	20.26 ± 0.04	11.72 ± 0.21
LG6	LSDV skin tissue	20.57 ± 0.17	18.72 ± 0.10
LG7	LSDV skin tissue	24.34 ± 0.01	17.56 ± 0.05
LG8	LSDV skin tissue	27.34 ± 0.04	11.37 ± 0.02
LG9	LSDV whole blood	14.62 ± 0.07	10.44 ± 0.25
LG10	LSDV whole blood	14.77 ± 0.17	12.36 ± 0.16
LG11	LSDV whole blood	15.30 ± 0.07	17.65 ± 0.04
LG12	LSDV whole blood	16.33 ± 0.12	11.56 ± 0.04
LG13	LSDV whole blood	16.60 ± 0.21	14.84 ± 0.14
LG14	LSDV whole blood	16.66 ± 0.03	10.07 ± 0.07
LG15	LSDV whole blood	16.83 ± 0.14	12.54 ± 0.15
LG16	LSDV whole blood	17.07 ± 0.07	13.52 ± 0.06
LG17	LSDV whole blood	17.07 ± 0.06	16.60 ± 0.06
LG18	LSDV whole blood	17.49 ± 0.05	10.65 ± 0.41
LG19	LSDV whole blood	17.73 ± 0.08	10.33 ± 0.14
LG20	LSDV whole blood	18.45 ± 0.03	12.83 ± 0.14
LG21	LSDV whole blood	19.77 ± 0.11	14.29 ± 0.09
LG22	LSDV whole blood	23.14 ± 0.13	32.69 ± 0.06
LG23	LSDV whole blood	24.83 ± 0.23	16.21 ± 0.06
LG24	LSDV whole blood	25.23 ± 0.01	11.40 ± 0.07
LG25	LSDV lymph node	16.82 ± 0.23	10.22 ± 0.04
LG26	LSDV lymph node	18.39 ± 0.07	16.29 ± 0.08
LG27	LSDV lymph node	19.59 ± 0.21	14.81 ± 0.14
LG28	LSDV lymph node	21.26 ± 0.04	20.45 ± 0.16
LG29	LSDV lymph node	21.46 ± 0.10	17.43 ± 0.11
LG30	LSDV lymph node	22.19 ± 0.08	16.25 ± 0.01
N1	Goat skin tissue	None	None
N2	Goat skin tissue	None	None
N3	Goat skin tissue	None	None
N4	Goat whole blood	None	None
N5	Goat whole blood	None	None
N6	Goat whole blood	None	None
N7	Goat whole blood	None	None
N8	Goat whole blood	None	None
N9	Goat whole blood	None	None
N10	Goat whole blood	None	None
N11	Goat whole blood	None	None
N12	Goat whole blood	None	None
N13	Goat lymph node	None	None
N14	Goat lymph node	None	None
N15	Goat lymph node	None	None
N16	Goat lymph node	None	None
N17	Sheep skin tissue	None	None
N18	Sheep skin tissue	None	None
N19	Sheep whole blood	None	None
N20	Sheep whole blood	None	None
N21	Sheep whole blood	None	None
N22	Sheep whole blood	None	None
N23	Sheep whole blood	None	None
N24	Sheep whole blood	None	None
N25	Sheep whole blood	None	None
N26	Sheep whole blood	None	None
N27	Sheep whole blood	None	None
N28	Sheep whole blood	None	None
N29	Sheep whole blood	None	None
N30	Sheep lymph node	None	None
N31	Sheep lymph node	None	None
N32	Cattle skin tissue	None	None
N33	Cattle skin tissue	None	None
N34	Cattle skin tissue	None	None
N35	Cattle skin tissue	None	None
N36	Cattle whole blood	None	None
N37	Cattle whole blood	None	None
N38	Cattle whole blood	None	None
N39	Cattle whole blood	None	None
N40	Cattle whole blood	None	None
N41	Cattle whole blood	None	None
N42	Cattle whole blood	None	None
N43	Cattle whole blood	None	None
N44	Cattle whole blood	None	None
N45	Cattle whole blood	None	None
N46	Cattle whole blood	None	None
N47	Cattle whole blood	None	None
N48	Cattle whole blood	None	None
N49	Cattle lymph node	None	None
N50	Cattle lymph node	None	None

**Table 6 microorganisms-11-02476-t006:** Diagnostic performance comparison between real-time qPCR and WOAH SYBR Green PCR.

Assays	Result	WOAH SYBR Green PCR			Performance Characteristics (%)		Agreement *Kappa* Value
		Positive	Negative	Total	Sensitivity	Specificity	
**Real-time qPCR**	Positive	85	0	85	100% (95.8–100%, 95% CI)	100% (92.9–100%, 95% CI)	1.0 (1-1, 95% CI)
	Negative	0	50	50			
	Total	85	50	135			

## Data Availability

All datasets generated for this study are included in the article.

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
