# Peer review of "Development of a Real-Time qPCR Method for the Clinical Sample Detection of Capripox Virus"

_microorganisms, 2023, doi:10.3390/microorganisms11102476_

Round 1
Reviewer 1 Report
The authors report a method for detection of Capripox Virus by TaqMan probes. There seems to be little problem with the validation items and the content.
One point I questioned is that the sample extraction method must be optimized for each infectious disease of interest to be detected by PCR. In Table 4, it is explained that various infectious diseases, such as foot-and-mouth disease and Peste-des-petits-ruminants virus, were not detected by the method developed in this study. However, not only those detected in serum, but also those that are intracellular parasites, such as Mycobacterium tuberculosis, and those, such as Bacillus, for which DNA is unlikely to be detected in the supernatant because it would collect toward sedimentation if centrifuged in the saline. DNA samples that are PCR-positive by the usual method for each infection should be used to clearly demonstrate whether they were negative by the PCR method developed in this study.
A minor point is the poor resolution of the figure in Figure 1; please use a higher resolution figure when submitting or printing.
Reviewer 2 Report
The research was devoted to the development of quantitative real time PCR in order to detect capripoxviruses. Real time PCR systems with fluorescent hydrolysis probe and double strand DNA intercalating fluorescent dye SYBR Green were compared. Undoubtly, the topic is of interest, nice experiments are shown in excellent illustrations. However, several issues have to be addressed to improve the manuscript before considering it for publication in "Microorganisms" https://www.mdpi.com/journal/microorganisms
General comments
All viruses belong to one phylum Vira with two subphyla Deoxyvira and Ribovira for DNA and RNA containing viruses. Strictly speaking, viruses are not microorganisms since their sizes near 10 nm=0.1 um and they do not have ribosomes. Probably, the journal "Microorganisms" is not an optimal choice, isn't it?
There is a strict international patent to use TaqMan probes. Without special permission it is recommended to write "fluorescent hydrolysis probe" in order to avoid some possible technical problems.
Abbreviation list is absent and some specific uncommon acronyms are not used properly. Some of them are not introduced at the first mention in the text (for example, in Abstract or Introduction but later in the text).
Unfortunately, conclusion is missing. It should focus the attention on the main achievements.
Specific comments
Promlem points are highlighted in yellow in the attached file.
Abstact
Line 18. It remains unclear what "three viruses" were meant since they were not described before.
Line 23. What plasmid was used for positive control? What virus-specific fragment was cloned in unknown vector?
Line 30. Abbreviation "OIE" was not previously described.
Introduction
Epidemiology of CaP viruses was not described. The only general statement "seriously prevalent" without exact data in different isolated populations with current trends of growth or decline.
Lines 77-78. "Laboratory diagnosis mainly relies on molecular biology and immunological detection methods"
For comprehensive comparison with the following research results it seems reasonable to specify all commercially available and previously described in scientific publications CaPV detection systems with their specificity and sensitivity levels.
Materials and Methods
Viruses from "clinical samples" of unknown origin are not enough for their characterization.
Time and place of their collection, natural and laboratory hosts, numbers in a collection where they are currently stored as well as GenBank accession numbers to confirm their specificity in addition to unidentified "sheep (or goat) infection" (Table 1) must be included.
Table 1. GPV, SPV and LSDV are serotypes or three separate species of CaPV?
Line 107. Details of homogenization procedure are required.
What means "normal saline"? What salt(s), what concentration(s)?
Line 108.
10,000 g is not high-speed centrifugation.
Line 115, 118. Multiple magnetic beadS are used in kits for DNA isolation, aren't they?
Line 128. What criteria were used for selection of certain "members" for multiple alignment of nucleotide sequences? Are "members" the natural isolates, strains, serotypes or species?
Lines 140-143. "Sequences were sequenced"? Probably, multiple alignment?
New original nucleotide sequences should be submitted in GenBank. What software was used for the alighnment of the nucleotide sequences?
Line 152. Fluorescein derivatives are not stable and their fluorescence emission depends on pH, protein concentrations and presence of pyrine nucleotide residues in a close proximity to the fluorophore. Rhodamine and cyanine fluorescent dyes are better.
Line 154. Terminal 5'- and 3'-localization of FAM and BHQ1 results in high background fluorescence (threshold level). Position of quenchers "IN" not far from fluorophores is desirable to detect weak, latent or persistent infections.
Table 3. Size of the PCR product is 209 bp according to positions of the forward and reverse primers but not 199 bp as shown in upper line with specific probe?
Line 153. Origin of "550 bp sequence of CaPV P32 gene" remains unclear. Other fragments corresponding to specific PCR products were described in Table 3.
Results.
Figure 1. GenBank Accession numbers and even "txt" do not permit to identify representatives of outgroups. Time and place of collections of CaPV isolates might increase the significance of the alignment.
Figure 3. Concentrations are shown the amounts of copies or ng in a volume (1 L, 1 ml, 1 ul).
Real concentrations or copies number in a reaction mixture are shown?
Figure 4. There are more than 3 fluorescent curves on the figure 4 for 3 CaPV (GPV, SPV and LSDV). All others are supposed to be negative?
Lines 341 and 343 are marked to demonstrate a wrong style.
Table 5. Two columns with so called "results" that show "+" and "-" seem to be excessive. Average Ct values with standard deviations are more than enough.
Discussion.
Line 395. Standard NY/T 576-2015 is evident in China but not in other countries. Perhaps some details in comparison with foreign standards might be appropriate.
Conclusiion is required.

English editing and corrections are required.
